# COVID-19 Vaccines Status, Acceptance and Hesitancy among Maintenance Hemodialysis Patients: A Cross-Sectional Study and the Implications for Pakistan and Beyond

**DOI:** 10.3390/vaccines11050904

**Published:** 2023-04-27

**Authors:** Zara Amjad, Iqra Maryam, Maria Munir, Muhammad Salman, Mohamed A. Baraka, Zia Ul Mustafa, Yusra Habib Khan, Tauqeer Hussain Mallhi, Syed Shahzad Hasan, Johanna C. Meyer, Brian Godman

**Affiliations:** 1Department of Paediatrics, District Head Quarter (DHQ), Bhakkar 30000, Pakistan; zaramjad52@gmail.com; 2Department of Medicine, Jinnah Hospital Lahore, Lahore 54000, Pakistan; dr.iqramaryam@gmail.com; 3Department of Medicine, Faisalabad Medical University, Faisalabad 38000, Pakistan; mariamunir08@gmail.com; 4Institute of Pharmacy, Faculty of Pharmaceutical and Allied Health Sciences, Lahore College for Women University, Lahore 54000, Pakistan; msk5012@gmail.com; 5Clinical Pharmacy Program, College of Pharmacy, Al Ain Campus, Al Ain University, Abu Dhabi P.O. Box 64141, United Arab Emirates; mohamed.baraka2020@gmail.com; 6Clinical Pharmacy Department, College of Pharmacy, Al-Azhar University, Cairo 11651, Egypt; 7Discipline of Clinical Pharmacy, School of Pharmaceutical Sciences, Universiti Sains Malaysia, Gelugor 11800, Penang, Malaysia; 8Department of Pharmacy Services, District Headquarter (DHQ) Hospital, Pakpattan 57400, Pakistan; 9Department of Clinical Pharmacy, College of Pharmacy, Jouf University, Sakaka 72388, Saudi Arabia; 10Department of Pharmacy, School of Applied Sciences, University of Huddersfield, Huddersfield HD1 3DH, UK; 11Department of Public Health Pharmacy and Management, School of Pharmacy, Sefako Makgatho Health Sciences University, Ga-Rankuwa 0208, South Africa; 12South African Vaccination and Immunisation Centre, Sefako Makgatho Health Sciences University, Molotlegi Street, Ga-Rankuwa 0208, South Africa; 13Centre of Medical and Bio-Allied Health Sciences Research, Ajman University, Ajman P.O. Box 346, United Arab Emirates; 14Strathclyde Institute of Pharmacy and Biomedical Science (SIPBS), University of Strathclyde, Glasgow G4 0RE, UK

**Keywords:** COVID-19 vaccines, awareness, high-risk, hemodialysis, acceptance, hesitancy, Pakistan

## Abstract

COVID-19 vaccine hesitancy continues to be a widespread problem in Pakistan due to various conspiracy beliefs, myths and misconceptions. Since the hemodialysis population is at a higher risk of contracting infections, we sought to investigate the current COVID-19 immunization status and reasons for any vaccine hesitancy among these patients in Pakistan. This cross-sectional study was conducted among maintenance hemodialysis patients at six hospitals in the Punjab Province of Pakistan. Data were collected anonymously using a questionnaire. A total of 399 hemodialysis patients took part in the survey, the majority of them were male (56%) and aged 45–64 years. A calculated 62.4% of the patients reported receiving at least one dose of the COVID-19 vaccine. Of those vaccinated (249), 73.5% had received two doses and 16.9% had received a booster dose. The most common reasons for vaccination were “being aware they were at high risk” (89.6%), “fear of getting infected” (89.2%) and “willingness to fight against COVID-19-pandemic” (83.9%). Of the 150 patients who had not yet been vaccinated, only 10 showed a willingness to take the COVID-19 vaccine. The major reasons for refusal included “COVID-19 is not a real problem” (75%), the “corona vaccine is a conspiracy (72.1%)” and “I don’t need the vaccine” (60.7%). Our study revealed that only 62% patients receiving hemodialysis were partially or completely vaccinated against COVID-19. Consequently, there is a need to initiate aggressive approaches to educate this high-risk population in order to address their concerns with vaccine safety and efficacy as well as correct current myths and misconceptions to improve the COVID-19 immunization status in this population.

## 1. Introduction

Coronavirus disease (COVID-19), alongside public health measures to reduce its spread, has placed a considerable social, economic and health burden on society and patients since its emergence in 2019 [1,2,3,4,5,6]. Incomes across countries were appreciably affected by lockdown and other measures in the absence of effective treatments and vaccines, which included a slow-down or closure of businesses [7]. According to a report issued by the United Nations, COVID-19 pushed 34 million people into extreme poverty and resulted in a 3.2% reduction in the global economy, which is a major burden for low–middle income countries (LMICs). By the end of 2030, more than 130 million people will likely be living in extreme poverty as a consequence of the pandemic [1]. This included Pakistan with a projected 33.7% increase in poverty as a result of social distancing and lockdown measures including restrictions on travel [8]. This is alongside to the general economic burden in Pakistan resulting from COVID-19 and the associated public health measures [9]. The healthcare system of almost every country, including Pakistan, was under extreme pressure initially with the disruption of essential supplies and a lack of diagnostics and treatment facilities. This was in addition to the mental and physical exhaustion of healthcare professionals alongside the morbidity and mortality associated with COVID-19 [10,11,12].

In Pakistan, the first case of COVID-19 was reported on 26 February, 2020, and after that, the country faced different disease waves despite adopting various preventive measures [13,14]. The first wave of COVID-19 in Pakistan peaked in mid-June 2020, infecting more than 300,000 people [15,16]. However, the government of Pakistan rapidly introduced preventive measures to limit the spread of the virus. These included strict lockdown activities, travel restrictions and enhanced capacity testing for COVID-19 as well as improved contact tracing and isolation in the absence of vaccines and effective treatments, which helped to reduce the number of positive cases [15,17,18,19]. The second wave of COVID-19 was declared by the government of Pakistan in October 2020, with greater infection, positivity and mortality rates [20,21]. The third COVID-19 wave impacted mainly the Punjab and Khyber Pakhtunkhwa provinces of Pakistan, with the peak of cases in this wave occurring in April 2021 [22,23]. The fourth wave of COVID-19 was declared in Pakistan in July 2021, with a positivity rate of 6.78% [24]. According to the National Command and Operation Center (NCOC), Government of Pakistan, the omicron-driven fifth COVID-19 wave peaked on 23 January 2022 [25]. Currently, the country is under the influence of a sixth COVID-19 wave, with more than 1,580,327 confirmed cases and 30,654 deaths reported in Pakistan as of 16 April 2023 [26]. On 16 April 2023, 1,547,909 patients had recovered, with 70 positive cases reported in the last 24 h. [27]. Most of the cases manifested asymptomatic to mild to moderate disease, with severe or critical cases hospitalized similar to previous waves [28,29,30].

Encouragingly, previous studies from Pakistan have indicated that the general population, university students and healthcare providers possessed adequate disease awareness and attitudes and proactive practices against COVID-19 [31,32,33,34]. Moreover, healthcare providers serving in COVID-19 health facilities appeared to have adequate adherence to infection control and preventive practices despite limited facilities [35].

The lack of effective treatments, despite many being proposed, along with the substantial economic and health consequences of the pandemic, resulted in considerable activities among scientists to develop safe and effective vaccines against COVID-19 [36,37,38,39,40,41,42]. This is because vaccines were considered the best option to save lives and contain SARS-CoV-2 on a global scale [40,42,43,44]. Despite having a fragile healthcare delivery system with insufficient standard diagnostic and treatment facilities, the government of Pakistan decided to vaccinate its population free of charge assisted by the COVAX initiative [45]. In this context, COVID-19 vaccines were first administered to frontline healthcare workers and subsequently to senior citizens (above 60 years old) [46,47]. Currently, every citizen in Pakistan above 5 years old can receive COVID-19 vaccines, including first or second boosters, from established COVID-19 vaccine centers [48]. This has been facilitated by the government of Pakistan providing the vaccines free-of-charge; otherwise, the costs would have been prohibitive for an appreciable proportion of the population [45,49]. Employing all available resources, including the establishment of COVID-19 vaccine centers in every district and tehsil in the country, mobile COVID-19 vaccine services for every market, town and village and door-to-door campaigns, an estimated 75.5% of the population were fully vaccinated, with 78.0% of the population receiving at least one dose by the beginning of March 2023 [50].

Shrestha et al. (2021) recently estimated that the prevalence of chronic kidney disease (CKD) in the general population in Southeast Asian countries, including Pakistan, was 14% in the general adult population, higher in males than females [51]. Hasan et al. (2018) estimated that the overall prevalence of CKD in Pakistan was 23.3% [52], with Alam et al. (2014) documenting a rate of 10.5% among those below 30 years of age and rising to 43.6% among the elderly (age above 50 years) [53]. End-stage renal disease (ESRD) is the last stage of CKD and is defined as a condition in which current renal function cannot meet the homeostasis of waste, fluid and electrolytes. However, there are concerns with available funding, especially among LMICs, to effectively manage patients with ESRD [54]. This will be further exacerbated among LMICs by the economic consequences of COVID-19 and associated lockdown measures [1,9].

More than 2 million ESRD patients require dialysis or kidney transplantation worldwide, whereas hemodialysis is one of the most frequently used modalities among patients with ESRD [55]. Patients on maintenance hemodialysis (MHD) are at greater risk of mortality due to associated comorbidities. These include hypertension, diabetes mellitus, cardiovascular complications and an increased prevalence of life-threatening infections, ultimately leading to immunity suppression among this vulnerable population [56,57]. Patients with MHD need to visit designated dialysis centers one to three times a week. Unavoidable proximity to other patients and healthcare providers, however, increases the risk of infection transmission in them and is exacerbated by infectious diseases such as COVID-19 and the consequences.

In recent studies, it has been reported that the prevalence of COVID-19 was greater among patients with MHD (5.3% to 36.2%). This could be due to a number of factors including traveling to and from outpatients’ settings and inevitable contact with healthcare personnel unable to practice social distancing, which resulted in a higher mortality (20% more) than in the general population [58,59,60,61]. Consequently, patients at higher risk of dying from COVID-19, such as these patients, should be fully vaccinated. This is especially the case in these patients as we are aware that there is excessive prescribing of antibiotics in patients with COVID-19 despite limited evidence of bacterial infections or co-infections, which drive up resistance rates and the associated implications [14,62,63]. 

A previous study conducted among patients with MHD indicated that they had higher hepatitis B hesitancy, and only 19.9% were fully vaccinated at the time of the study [64]. This is a concern when it comes to vaccination against COVID-19 in this population. This is because, despite high vaccination rates in Pakistan, there are concerns with vaccine hesitancy among healthcare workers (5.2–21.5%) and the general population (21.5–49%) in Pakistan, similar to many other LMICs. This is due to concerns with the safety and effectiveness of the vaccines as well as misbelief and misinformation fueled by fake news [65,66,67,68,69,70,71,72]. Similar to other countries, this needs to be avoided especially in this highly vulnerable population to reduce future morbidity and mortality. This is similar to other high-risk populations and must be a focus for vaccination strategies across countries. To the best of our knowledge, no study has yet been conducted among patients with MHD from Pakistan regarding their COVID-19 vaccination status as well as key factors associated with acceptance or hesitancy. Consequently, we sought to address this information gap by conducting a multicenter study among patients with MHD in Punjab Province in Pakistan. Punjab Province was chosen for this study in view of its population size within Pakistan, i.e., accounting for more than half of the population of Pakistan [35,63,73].

## 2. Materials and Methods

### 2.1. Study Design, Population and Location

A cross-sectional, questionnaire-based survey was conducted among MHD patients in Punjab Province between April 2022 and June 2022. The government of Punjab has two separate ministries within the public sector health department. One is the ‘Specialized Healthcare & Medical Education Department’, which looks after all the tertiary care hospitals in the province, whereas the ‘Primary & Secondary Healthcare Department’ is the controlling authority of district headquarter hospitals (DHQHs), tehsil headquarter hospitals (THQHs), rural health centers (RHCs) and basic health units (BHU) [73].

Punjab Province is currently divided into ten major divisions, with each division having districts and tehsils. Each division has at least one tertiary care/teaching hospital, whereas DHQHs and THQHs are established in respective districts and tehsils for the provision of health services. Primary care hospitals, RHCs and BHUs are established in small towns and villages, respectively.

Hemodialysis facilities are currently provided at tertiary and secondary care hospitals (DHQH and THQH) in Punjab Province. All these hospitals are equipped with designated dialysis units, with appropriate medical staff and the necessary medicines and laboratory facilities free-of-charge.

A convenient sampling technique was employed to collect data from the hemodialysis population from six hospitals (one tertiary care, three DHQHs and two THQHs) in Punjab Province. Approximately 50–70 patients were enrolled in tertiary care hospitals, 40–55 in DHQHs and 15–25 in THQHs. These are all public sector hospitals as this is where the vast majority of patients with MHD are treated in the province, with care provided free-of-charge.

### 2.2. Study Instrument

The study questionnaire (Appendix A) was developed from previous surveys conducted by the co-authors on a similar topic and after receiving permission from the corresponding authors to use the questionnaire as a basis for this study [74,75,76,77,78,79,80]. The content validity of the questionnaire was assessed by a 7-member team comprising medical doctors and pharmacists. After incorporating the recommendations/suggestions from the team, a final draft of the study instrument was prepared. The internal consistency of the study instrument was accessed and fell within the acceptable range (Cronbach’s alpha ≥ 0.7). A pilot study was subsequently conducted prior to the initiation of the full-length study by inviting 20 potential participants. We have successfully used this approach in previous studies where there is no specific validated instrument; however, one can be successfully adapted from previous studies using the experience of the co-authors [81,82,83,84,85]. After incorporating the recommendations/suggestions of the participants in the pilot study, the study instrument had the following three sections:Section-I: This section collected data on the demographics of the study population, hemodialysis-related details and the COVID-19 vaccination status.Section-II: This section consisted of nine questions relating to the factors associated with the acceptance of COVID-19 vaccines among the study population. Each question had a ‘yes’ and ‘no’ option, and participants were requested to select one option for each question.Section-III: This section collected information on the reasons for COVID-19 vaccine hesitancy among study participants. There were twelve questions in this section and, similar to Section II, two response options were available, ‘yes’ and ‘no’, of which respondents had to select one option.

### 2.3. Sample Size Calculation

In Pakistan, there is no central registry or online database to document the total number of patients with MHD throughout the country. Consequently, we were unable to identify the total number of the study population. A sample size was subsequently computed by using Raosoft, 206-525-4025 (US), which is an online sample size calculator [86]. Assuming an expected frequency of 50%, 95% confidence interval and 5% margin of error, the minimum sample size was 376 hemodialysis patients.

### 2.4. Sampling and Data Collection Procedure

We invited the patients with MHD of the above-mentioned health facilities to participate in our study. Those who were willing to participate were included in our survey. The team of investigators comprised healthcare professionals (doctors, pharmacists, pharmacy technicians, dialysis technicians and nurses) based at the study facilities, who subsequently collected data from their health facilities. The investigators visited potential patients for inclusion in the study and briefed them about the objectives of the study prior to data collection. No personal information was collected, and written informed consent was obtained from all study respondents before participation in the study. Study participants were informed that they could terminate participation at any stage of data collection without providing any reason and that this would not affect their subsequent care.

### 2.5. Statistical Analysis

All data were analyzed using SPSS version 22. Categorical data were represented as frequencies and percentages, whereas means and standard deviations were calculated for all the continuous variables. Comparisons of characteristics between vaccinated and non-vaccinated patients were made using the Chi-Square or Fisher’s Exact test as appropriate. A 2-sided *p*-value of ≤0.05 was considered statistically significant.

### 2.6. Ethical Approval

The protocol was reviewed and approved by the Office of Research, Innovation and Commercialization (ORIC), Lahore College for Women University, Jail Road, Lahore (letter no. ORIC/LCWU/448A). Approvals were also sought from the administration of the hospitals involved in the study before commencement of the study. The confidentiality of participants was ensured by not collecting any personal information during the data collection procedure and keeping all data anonymous.

## 3. Results

A total of 399 hemodialysis patients participated in the survey. The characteristics of the study participants are summarized in Table 1 and distributed according to their COVID-19 vaccination status. Most patients were between 45 and 64 years of age (50.6%), followed by those above 65 years old (35.1%). More than half of the participants (56.4%) were males, and 62.2% resided in rural areas. The majority of the surveyed patients had a low family income and only 20.6% were employed. Only 22.3% had received the influenza vaccine, whereas 73.4% were vaccinated against hepatitis B. A sizeable proportion of the surveyed patients (40.6%) reported that they had a family member or a relative who had contracted COVID-19, and 5.5% reported having a family member and/or relative who died due to COVID-19.

The most common comorbidities among the study participants were hypertension (41.6%), followed by diabetes mellitus (20.6%).

### 3.1. COVID-19 Vaccination Status among MHD Patients

As shown in Table 1, 62.4% of patients (249/399) reported receiving at least one dose of the COVID-19 vaccine. Of these 249 patients, 73.5% (*n* = 183) had been administered two doses of the COVID-19 vaccine and 16.9% (*n* = 42) had received an additional booster (Figure 1).

### 3.2. Reasons for COVID-19 Vaccine Acceptance

Figure 2 contains the reasons for COVID-19 vaccine uptake in the study population. The three principal reasons included “being aware they were at high risk” (89.6%), “fear of getting infected” (89.2%) and “responsibility of their role in fighting the pandemic by getting themselves vaccinated” (83.9%).

Demographic comparisons between COVID-19 vaccinated and non-vaccinated patients undergoing hemodialysis are presented in Table 1. There was no statistically significant difference (*p* > 0.05) of any variable between vaccinated and unvaccinated patients.

### 3.3. Reasons for COVID-19 Vaccine Hesitancy

Of the 150 unvaccinated patients in our study, only 10 patients (6.7%) reported willingness to subsequently be vaccinated, whereas the remainder (*n* = 140, 93.3%) refused the vaccine. The reasons for vaccine refusal are illustrated in Figure 3. The three most common reasons for COVID-19 vaccine refusal were that COVID-19 was not believed to be a real problem (75.0%), the COVID-19 vaccine was believed to be a conspiracy (72.1%) and ‘I don’t think I need this vaccine’.

## 4. Discussion

We believe our study is one of the first exploratory studies conducted in a high-risk population in Pakistan to ascertain the current status of COVID-19 vaccinations among this crucial population. In addition, we evaluated the reasons for COVID-19 vaccine acceptance among those who had been vaccinated and the reasons for hesitancy amongst those who indicated that they were not prepared to be vaccinated. The latter group is particularly important to target with pertinent messaging, with similar messaging potentially useful to address concerns in other high-priority groups for vaccination. Our study revealed that approximately two-thirds of the MHD patients had been administered at least one COVID-19 vaccine. These findings are comparable to the study reported from the US, in which slightly more than half of MHD patients had received at least one dose of a COVID-19 vaccine [74]. In another study from Italy, most of the MHD patients (>95%) had received at least one vaccine dose against COVID-19 [75].

Encouragingly, around three-quarters (73.5%) of the vaccinated patients had received at least two doses of the COVID-19 vaccine, with 16.9% having received a booster dose as well. These findings are consistent with the findings of a UK study, in which most of the MHD patients had received two doses of COVID-19 vaccines [76].

Common reasons for COVID-19 acceptance in our study population were being a high-risk population as well as fears of contracting COVID-19 and its consequences. A previous study from Egypt highlighted similar motivators for COVID-19 vaccine acceptance among MHD patients [77]. Another common reason was the willingness of the MHD patients to fight against COVID-19. These findings are similar to a previous study from China, in which most of the study participants accepted COVID-19 vaccines due to the fact that they were ready to fight against the COVID-19 pandemic [87]. Other potential reasons for COVID-19 vaccine acceptance reported in our study were the fear of transmitting COVID-19 to others, the government making COVID-19 vaccination compulsory, not being afraid of possible side-effects of COVID-19 vaccines and the desire to return to normalcy as soon as possible. These findings are in contrast to an Egyptian study which concluded that a very small percentage of participants’ motivators for COVID-19 vaccines were the transmission of COVID-19 to others and not having a fear of any side-effects of COVID-19 vaccines [77]. We believe the potential reason for no fear of any possible side-effects associated with COVID-19 vaccines in our study was currently a very limited number of side-effects actually being reported by the Drug Regulatory Authority of Pakistan (DRAP) compared to other countries [78]. However, despite the fact that 64.3% of vaccinated MHD patients indicated that they were not afraid of the possible side-effects of the vaccine, only 41.8% indicated they had no doubts about the safety of the COVID-19 vaccines. This is important because the safety of the vaccines will be a key issue going forward [88,89,90].

Our study also revealed more than 37% of the study population was hesitant towards COVID-19 vaccination, with only a limited percentage of unvaccinated patients showing a willingness to receive the COVID-19 vaccine. This is a concern despite levels of COVID-19 vaccine hesitancy being quite similar across many LMICs including Pakistan [79,80,91]. As far as the factors associated with hesitancy towards the COVID-19 vaccines are concerned, our study revealed that three-quarters of the hesitant population stated that COVID-19 is not a real problem, followed by COVID-19 vaccines being considered a conspiracy. This is in line with the level of misinformation regarding conspiracy theories across countries including Pakistan [69,71,92,93]. In contrast to the findings of our study, MHD patients in the UK believed that COVID-19 was overhyped by the media, and they were not convinced by the benefits of COVID-19 vaccines [94]. Other reasons for COVID-19 vaccine hesitancy reported in our survey were ‘no need for COVID-19 vaccines’, ‘if other people are getting COVID-19 vaccines then I won’t need it’ and falsified information circulating in the media about the safety concerns of available COVID-19 vaccines. These reasons for hesitancy reported in our study are similar to other countries and in other serious conditions including cancer [95,96,97,98]. This needs to be addressed.

To address this, we believe the federal as well as provincial health authorities in Pakistan should implement appropriate interventions to enhance COVID-19 vaccine coverage amongst high-risk populations, including MHD patients. Potential ways forward include not automatically assuming that priority groups such as those with MHD or cancer will automatically wish to be vaccinated [98,99]. However, this can change over time with an increase in information [99]. In addition to increasing trust in governments, which has been eroded as a result of the pandemic and subsequent actions, developing and promoting community champions and leaders as well as proactively addressing misinformation and mistrust [99,100,101,102]. The latter needs to acknowledge the increasing role of social media, which up to now has been a key source of misinformation regarding COVID-19, possible treatments and vaccines [39,92,96,103,104]. The content of any social media information campaign for this and future pandemics needs to be constructed using easy-to-understand language and targeting key areas with fact-based information, which includes addressing concerns with the side-effect of vaccines with up-to-date information, with healthcare workers active in rapidly dispelling myths and misinformation [39,105,106,107]. National and regional health authorities must also facilitate vaccination for these high-risk people by adopting mobile vaccination techniques or the routine provision of COVID-19 vaccines at hemodialysis clinics free-of-charge. Moreover, technology-based health literacy demonstrations including mobile phone recall text messaging in local languages, pictorial messages, automated phone calls or interactive voice recording for spreading awareness of the risks of COVID-19 and the benefits of the vaccine can also help address current concerns in this high-risk group [108]. Similar strategies can also be adopted for other high-risk groups in Pakistan and beyond. Finally, it is important for current and future pandemics that integrated care pathways are developed for high-risk groups such as those with MHD to take account of their needs as well as their protection against infectious diseases [109]. We will be following this up in future studies.

We are aware that our study has a number of limitations. Firstly, we gathered data from MHD patients from only six hospitals in the Punjab Province. However, we chose Punjab Province for the reasons stated. In addition, we chose a range of hospitals in the Punjab Province to reflect the different hospital facilities where hemodialysis facilities are currently provided. Secondly, we included only public sector hospitals in the current study and did not include private sector hospitals for the reasons stated. Thirdly, we employed a non-probability sampling method to recruit hemodialysis patients, which has disadvantages including selection bias and non-generalizability. Lastly, we acknowledge that we have not tested possible educational and other approaches to address vaccine hesitancy in this population as this was not the aim of this paper. This will be followed up in future studies. Despite these limitations, we believe our findings are robust in providing direction to the future to all key stakeholders in Pakistan and beyond in seeking ways to improve vaccination rates in this high-risk group.

## 5. Conclusions

Our study concluded that 62.4% of MHD patients had been administered a COVID-19 vaccine, with a major reason for this being their awareness that they were at a high risk of becoming infected compared to the general population, with the associated impact on future morbidity and mortality. We believe this is likely to increase. However, it was of concern that the willingness to be vaccinated was extremely low among unvaccinated patients. A significant proportion of hemodialysis patients were hesitant towards vaccines due to conspiracy beliefs, myths and misconceptions regarding COVID-19 and its vaccines. Health authorities and other key stakeholder groups including healthcare professionals should take measures to address highlighted concerns in this high-risk population. This includes educational and other approaches incorporating social media and other channels.

## Figures and Tables

**Figure 1 vaccines-11-00904-f001:**
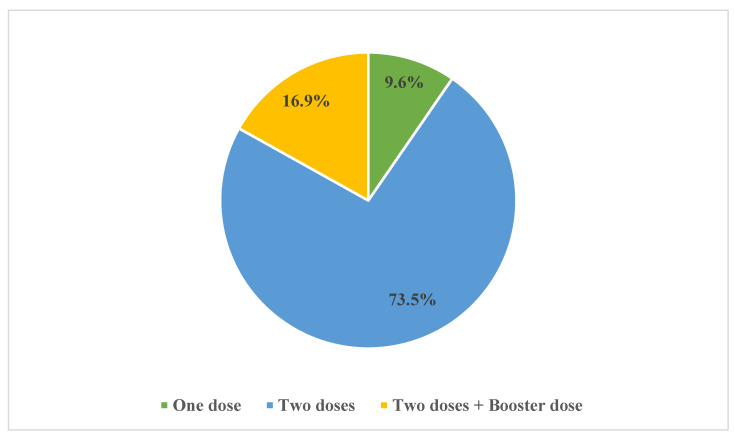
Distribution of the number of doses received amongst patients who had been vaccinated against COVID-19 (*n* = 249).

**Figure 2 vaccines-11-00904-f002:**
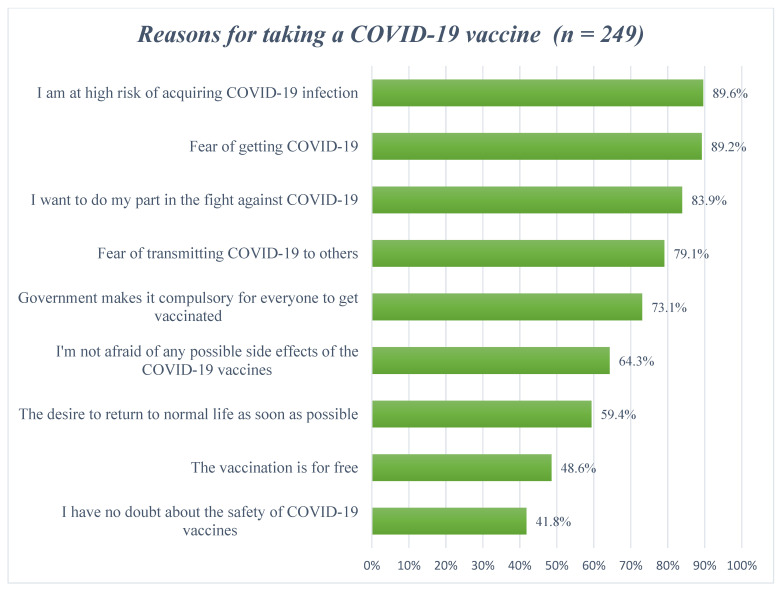
Reasons for COVID-19 vaccine acceptance ranked according to percentage agreement.

**Figure 3 vaccines-11-00904-f003:**
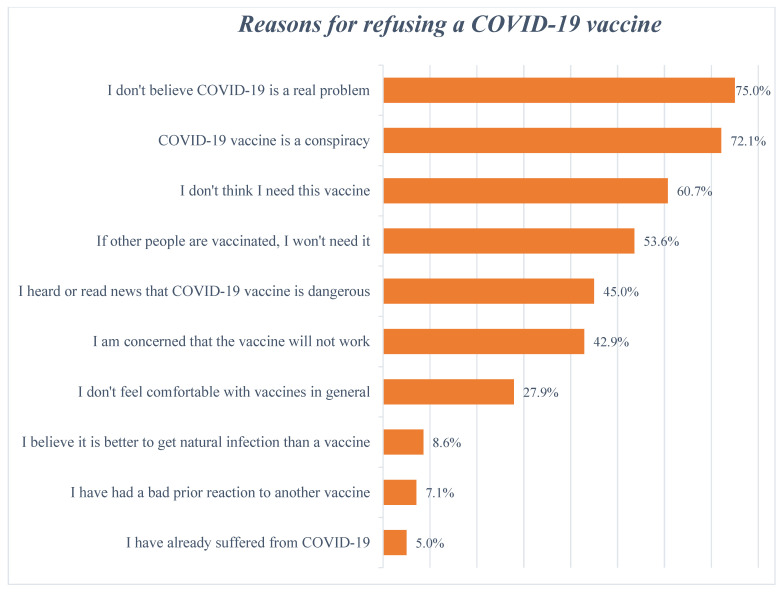
Reasons for COVID-19 vaccine refusal.

**Table 1 vaccines-11-00904-t001:** Characteristics of COVID-19 vaccinated and unvaccinated hemodialysis patients.

Variable	Sub-Groups	N (%)			*p*-Value
		Overall (*n* = 399)	Vaccinated (*n* = 249)	Unvaccinated (*n* = 150)	
**Age (years)**	18–44	57 (14.3)	32 (12.9)	25 (16.7)	0.563
	45–64	202 (50.6)	129 (51.8)	73 (48.7)	
	≥65	140 (35.1)	88 (35.3)	52 (34.7)	
**Gender**	Male	225 (56.4)	138 (55.4)	87 (58.0)	0.677 *
	Female	174 (43.6)	111 (44.6)	63 (42.0)	
**Residence**	Urban	151 (37.8)	94 (37.8)	57 (38.0)	1.000
	Rural	248 (62.2)	155 (62.2)	93 (62.0)	
**Marital status**	Single	111 (27.8)	61 (24.5)	50 (33.3)	0.065
	Married	288 (72.2)	188 (75.5)	100 (66.7)	
**Occupation**	Employed	82 (20.6)	55 (22.1)	27 (18.0)	0.501
	Unemployed	258 (64.7)	159 (63.9)	99 (66.0)	
	Retired	42 (10.5)	23 (9.2)	19 (12.7)	
	Cannot work due to disability	17 (4.3)	12 (4.8)	5 (3.3)	
**Family income (PKR)**	<30,000	182 (45.6)	108 (43.4)	74 (49.3)	0.498
	31,000–60,000	129 (32.3)	83 (33.3)	46 (30.7)	
	>60,000	88 (22.1)	58 (23.3)	30 (20.0)	
**Education**	Illiterate	104 (26.1)	63 (25.3)	41 (27.3)	0.290
	Religious education only	33 (8.3)	24 (9.6)	9 (6.0)	
	Primary	88 (22.1)	55 (22.1)	33 (22.0)	
	Secondary	105 (26.3)	65 (26.1)	40 (26.7)	
	High secondary	46 (11.5)	24 (9.6)	22 (14.7)	
	Diploma/Bachelor/Master	23 (5.8)	18 (7.2)	5 (3.3)	
**Smoking status**	Current smoker	79 (19.8)	51 (20.5)	28 (18.7)	0.907
	Non-smoker	233 (58.4)	144 (57.8)	89 (59.3)	
	Former smoker	87 (21.8)	54 (21.7)	33 (22.0)	
**Years on hemodialysis**	<1	59 (14.8)	37 (14.9)	22 (14.7)	0.915
	>1 to 3	181 (45.4)	111 (44.6)	70 (46.7)	
	>3	159 (39.8)	101 (40.6)	58 (38.7)	
**Frequency of hemodialysis**	Once a week	8 (2.0)	4 (1.6)	4 (2.7)	0.539
	Two times a week	310 (77.7)	191 (76.7)	119 (79.3)	
	Three times a week	81 (20.3)	54 (21.7)	27 (18.0)	
**Influenza vaccine**	Vaccinated	89 (22.3)	57 (22.9)	32 (21.3)	0.804 *
	Not vaccinated	310 (77.7)	192 (77.1)	118 (78.7)	
**Hepatitis B vaccine**	Vaccinated	293 (73.4)	184 (73.9)	109 (72.7)	0.815 *
	Not vaccinated	106 (26.6)	65 (26.1)	41 (27.3)	
**Hepatitis C**	Positive	106 (26.6)	64 (25.7)	42 (28.0)	0.641 *
	Negative	293 (73.4)	185 (74.3)	108 (72.0)	
**Family member or relative infected with COVID-19**	Yes	162 (40.6)	106 (42.6)	56 (37.7)	0.344 *
	No	237 (59.4)	143 (57.4)	94 (62.7)	
**Family member or relative died due to COVID-19**	Yes	22 (5.5)	15 (6.0)	7 (4.7)	0.655 *
	No	377 (94.5)	234 (94.0)	143 (95.3)	

* Fisher’s exact test.

## Data Availability

Additional data are available from the corresponding authors on reasonable request.

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
