# Peer review of "COVID-19 Vaccines Status, Acceptance and Hesitancy among Maintenance Hemodialysis Patients: A Cross-Sectional Study and the Implications for Pakistan and Beyond"

_vaccines, 2023, doi:10.3390/vaccines11050904_

Round 1

Reviewer 1 Report

The aim of this study is to understand the reason for vaccine hesitancy in hemodialysis patients living at the Punjab Province, Pakistan. The rationale of the study, considering that there is considerable evidence about this topic in different populations, including dialysis patients, is weak. This could be improved if more evidence is given about the special characteristics of Pakistan population in general and from its dialysis patients in particular.  

Specific comments:

·      Introduction: 1st paragraph is not related to the main topic of the study; at this point there is no need to add more information regarding COVID-19 disease and its social, economic and health burden. If this paragraph is reduced, I suggest giving more information on vaccine uptake and vaccine hesitancy in Pakistan (not only for COVID-19) or expanding the knowledge on hemodialysis patients in Pakistan and vaccination in this specific population. This could improve the novelty of the study.

·      Line 100: instead of multiple times a week, I suggest 1-3 times a week.

·      Lines 106-107: “there are concerns with vaccine hesitancy among health care workers and the general population of Pakistan”; this is a nonspecific comment; could you give information regarding vaccine hesitancy in these population in Pakistan?

·      Line 177: regarding the age of the participants (50.6% between 45-64), is this age range representative of hemodialysis patients in Pakistan?

·      Lines 191-193: At some point it will be important to give the vaccination status for general population in that region at the time of the study. Are these patients different with this respect?

·      Lines 231-233: “We believe our study is one of the first exploratory studies conducted in a high-risk population in Pakistan to ascertain the current status of COVID-19 vaccinations among this population”; this was indeed one of the mail objectives of the study, as mentioned in the introduction. If this was something important to know, is there any other reliable way to obtain this information (ie, public data on vaccine status in different population in the country?). 

·      Lines 299, considering the probability that the percentage of patients that receive the vaccine may increase during time, I will mention again in the conclusion the timing of the study.

Author Response

Comments and Suggestions for Authors

The aim of this study is to understand the reason for vaccine hesitancy in hemodialysis patients living at the Punjab Province, Pakistan. The rationale of the study, considering that there is considerable evidence about this topic in different populations, including dialysis patients, is weak. This could be improved if more evidence is given about the special characteristics of Pakistan population in general and from its dialysis patients in particular.  

Author comments: Thank you for this comment. We have now added in more information into the Introduction and Discussion as to the rationale behind this study (for Pakistan and beyond) including ways forward to target hesitancy in this high-risk group. We hope this is now acceptable. 

Specific comments:

1) Introduction: 1st paragraph is not related to the main topic of the study; at this point there is no need to add more information regarding COVID-19 disease and its social, economic and health burden. If this paragraph is reduced, I suggest giving more information on vaccine uptake and vaccine hesitancy in Pakistan (not only for COVID-19) or expanding the knowledge on hemodialysis patients in Pakistan and vaccination in this specific population. This could improve the novelty of the study.

Author comments: Thank you for this comment. We would like to keep paragraph one (in line with comments from other Reviewers and the Editor) as we believe this helps sets the scene for the study. We have subsequently added in more details about the waves of COVID-19 in Pakistan and their impact as well as current rates and concerns with vaccine hesitancy in Pakistan as a prelude to justifying this study in this high-risk group (as requested by other Reviewers and the Editor). We hope this is now OK.

2) Line 100: instead of multiple times a week, I suggest 1-3 times a week.

Author comments: Thank you – now changed.

3) Lines 106-107: “there are concerns with vaccine hesitancy among health care workers and the general population of Pakistan”; this is a nonspecific comment; could you give information regarding vaccine hesitancy in these population in Pakistan?

Author comments: Thank you. We have now added percentages of general population and health care workers in Pakistan that were hesitant towards COVID-19 vaccines, and hope this is now acceptable.

4) Line 177: regarding the age of the participants (50.6% between 45-64), is this age range representative of hemodialysis patients in Pakistan?

Author comments: Thank you for the inquiry. There is no central registry to document the demographics characteristics of hemodialysis patients. Based on the findings of our study as well study conducted by Amjad et al (new ref 64) we can say that majority of the study population were from 45-64 years age group. We hope this is now OK.

5) Lines 191-193: At some point it will be important to give the vaccination status for general population in that region at the time of the study. Are these patients different with this respect?

Author comments: Thank you for the comment. We are unable to find the vaccination status of general population in the region where we conducted this study; however, we have included the overall vaccination status in Pakistan (lines 108 - 112). We hope this is now acceptable.

6) Lines 231-233: “We believe our study is one of the first exploratory studies conducted in a high-risk population in Pakistan to ascertain the current status of COVID-19 vaccinations among this population”; this was indeed one of the mail objectives of the study, as mentioned in the introduction. If this was something important to know, is there any other reliable way to obtain this information (ie, public data on vaccine status in different population in the country?). 

Author comments: Thank you for this comment. We search through Pub Med and other databases and could not find any such studies – hence our comment. We hope this is acceptable.

7) Lines 299, considering the probability that the percentage of patients that receive the vaccine may increase during time, I will mention again in the conclusion the timing of the study.

Author comments. Thank you, now added

Reviewer 2 Report

The manuscript is interesting, the writing is easy to understand and very pleasant to read.

Ms has some limitations. Despite these limitations, it is possible to perceive that the results presented in the MS have great relevance and scientific importance. Providing guidance for the future in finding ways to improve vaccination rates in this high-risk group. I can say that the main objective proposed by the authors was elegantly achieved.

The authors concluded that 62.4% of patients with MHD who received a vaccine against COVID-19, the main reason for this being the awareness that they were at high risk of becoming infected compared to the general population, with the associated impact in future morbidity and mortality. The main reason for concern was that the interest in being vaccinated was virtually non-existent among patients who did not receive any dose of immunization.

It was described that a significant proportion of patients on hemodialysis hesitated about vaccines due to conspiracy beliefs, myths, misconceptions and especially fake news regarding COVID-19 and especially immunizations. Health authorities should take steps to address the concerns highlighted in this high-risk population, including educational approaches.

The manuscript is important mainly for highlighting the importance of immunization for COVID-19 and for other diseases that are easily controlled with the use of vacci

Author Response

Comments and Suggestions for Authors

1) The manuscript is interesting; the writing is easy to understand and very pleasant to read.

Ms has some limitations. Despite these limitations, it is possible to perceive that the results presented in the MS have great relevance and scientific importance. Providing guidance for the future in finding ways to improve vaccination rates in this high-risk group. I can say that the main objective proposed by the authors was elegantly achieved.

Author comments: Thank you for these kind words – appreciated!

2) The authors concluded that 62.4% of patients with MHD who received a vaccine against COVID-19, the main reason for this being the awareness that they were at high risk of becoming infected compared to the general population, with the associated impact in future morbidity and mortality. The main reason for concern was that the interest in being vaccinated was virtually non-existent among patients who did not receive any dose of immunization.

Author comment: Thank you for the comment. Our study revealed that the majority of the participants who hadn’t yet received their COVID-19 vaccines were hesitant due to a number of reasons. These included they didn’t consider COVID-19 was a real problem, COVID-19 was a conspiracy, and their belief that they didn’t need these vaccines. This needs to be addressed with suggestions going forward, which are included in the updated Discussion. We hope this is now acceptable.

3) It was described that a significant proportion of patients on haemodialysis hesitated about vaccines due to conspiracy beliefs, myths, misconceptions and especially fake news regarding COVID-19 and especially immunizations. Health authorities should take steps to address the concerns highlighted in this high-risk population, including educational approaches.

Author comments; You are correct. We have suggested certain approaches to be adopted by national and regional health authorities, as well as HCPs, to overcome vaccines hesitancy in high-risk groups such as these patients. Social media also needs to be used more by key stakeholder groups including health authority personnel in the future to address ongoing misinformation. We will continue to monitor this.

4) The manuscript is important mainly for highlighting the importance of immunization for COVID-19 and for other diseases that are easily controlled with the use of vaccines

Author comments: Thank you for this – appreciated.

Reviewer 3 Report

Dear Editor,

I am sharing my review of the Manuscript ID vaccines-2324513 entitled: Covid-19 vaccines status, acceptance and hesitancy among 2 maintenance hemodialysis patients: A Cross-sectional study. The study was conducted among hemodialysis patients in Pakistan to investigate their COVID-19 vaccination status and reasons for vaccine hesitancy. Out of the 399 participants, 62.4% reported receiving at least one dose of the vaccine, while 150 patients had not yet been vaccinated, with the majority citing reasons such as COVID-19 being a conspiracy or not a real problem. The study highlights the need for aggressive education to correct myths and misconceptions and improve vaccination rates among this high-risk population. The manuscript should be accepted after minor revision.

The following issues should be addressed:

An ethics number from the local ethics committee is missing; please add.

An image of the questionnaire might be helpful in the MM section.

Best

Author Response

Comments and Suggestions for Authors

I am sharing my review of the Manuscript ID vaccines-2324513 entitled: Covid-19 vaccines status, acceptance and hesitancy among 2 maintenance hemodialysis patients: A Cross-sectional study. The study was conducted among hemodialysis patients in Pakistan to investigate their COVID-19 vaccination status and reasons for vaccine hesitancy. Out of the 399 participants, 62.4% reported receiving at least one dose of the vaccine, while 150 patients had not yet been vaccinated, with the majority citing reasons such as COVID-19 being a conspiracy or not a real problem. The study highlights the need for aggressive education to correct myths and misconceptions and improve vaccination rates among this high-risk population. The manuscript should be accepted after minor revision.

Author comments: Thank you for your summary – appreciated! We hope we have adequately addressed the comments you made.

The following issues should be addressed:

1) An ethics number from the local ethics committee is missing; please add.

Author comments: Thank you for this. We had included this data at the end of the manuscript (in line with the requirements of the Journal). However – also now duplicated in Section 2. We hope this is now acceptable.

2) An image of the questionnaire might be helpful in the MM section.

Author comments: Thank you – now included under Supplementary Material with a URL link. We hope this is now OK.

Round 2

Reviewer 1 Report

The authors have adequately responded to my concerns.